# Assessment of Cervicovaginal Smear and HPV DNA Co-Test for Cervical Cancer Screening: Implications for Diagnosis and Follow-Up Strategies

**DOI:** 10.3390/diagnostics14060611

**Published:** 2024-03-13

**Authors:** Neslihan Kaya Terzi, Ozden Yulek

**Affiliations:** Department of Pathology, Faculty of Medicine, Canakkale Onsekiz Mart University, 17000 Canakkale, Turkey; ozdenuctu@hotmail.com

**Keywords:** cervical cancer screening, colposcopy, conventional smear, liquid-based cytology, HPV

## Abstract

Objective: Cervical cancer is a major cause of cancer-related mortality, necessitating effective screening and diagnostic methods. This study aimed to assess the performance of cervicovaginal smear (CVS) and human papillomavirus (HPV)-DNA co-test. Study design: The pathology results of 225 female patients who underwent HPV-DNA testing with CVS between 2014 and 2022 and were subsequently diagnosed by colposcopic cervical biopsy or second CVS were retrospectively analyzed. Results: CVS samples showed atypical squamous cells of undetermined significance (ASCUS), low-grade squamous intraepithelial lesion (LSIL), high-grade squamous intraepithelial lesion (HSIL), and cervical cancer. Concordance between the first and second diagnoses demonstrated moderate agreement for LSIL. ASCUS cases exhibited a significant correlation with HPV-DNA positivity and higher-grade cervical lesions. In biopsy, sensitivity and specificity for CIN1/LSIL were 87.5% and 53.3, respectively, while for CIN2-3/HSIL, they were 83.87% and 58.49%. HPV testing showed significant correlation with histopathologic results. In women over 40 years, more intraepithelial lesions were diagnosed compared to younger women (*p* < 0.005). The conventional smear technique proved reliable in detecting high-grade lesions. Conclusions: Despite the limitations of our study, our results emphasize the value of HPV-DNA testing to avoid unnecessary interventions and to establish appropriate follow-up strategies.

## 1. Introduction

Cervical cancer, the fourth most common cancer in women and the second leading cause of mortality among malignancies, resulted in 528,000 cases and 311,000 deaths worldwide in 2018. This epidemiologic profile underscores the cancer’s continued status as a major public health problem [1,2,3].

There are significant regional variations in incidence depending on the status of human papillomavirus (HPV) vaccination and the availability of cervicovaginal smear (CVS) screening [4,5].

In 2012, the American Society for Colposcopy and Cervical Pathology (ASCCP) revised a comprehensive, evidence-based consensus on the management of abnormal cervical cytology, cervical intraepithelial neoplasia, and adenocarcinoma. The ASCCP recommends screening with co-test HPV DNA and smear for women aged 30–64 years. This strategy not only achieves the highest sensitivity for cancer detection, but also exhibits the highest negative predictive value by providing longer intervals between negative test results [6,7].

In Turkish studies, ASCUS is the most common abnormal smear result, sparking controversy in management. Research heavily focuses on ASCUS and CIN1/LSIL patients, with ASCUS cases benefiting from comprehensive evaluation, preferably with HPV testing, especially with liquid-based cytology. Conversely, colposcopy is recommended for CIN1/LSIL patients, particularly with high HPV positivity. However, cytologic follow-up has drawbacks like missed CIN2-3/HSIL lesions, potential delays in cancer diagnosis, repeated testing, patient non-compliance, and increased anxiety [8].

Aydogan et al. conducted a study assessing cervical cytology using conventional methods. They found a sensitivity of 70.8% and specificity of 62.2% (with positive predictive value (PPV) and negative predictive value (NPV) both at 66.7%) for detecting low-grade lesions. Similarly, for high-grade lesions, the sensitivity was 72.4% and specificity was 86.0% (with PPV at 70.0% and NPV at 87.3%). Among patients with CIN2-3/HSIL or ASC-H, 73.5% were diagnosed with high-grade lesions following colposcopy. Moreover, CIN2/HSIL lesions were identified in 26.9% of patients with ASCUS and high-risk HPV. The results of smear-colposcopy-guided biopsy closely correlated with smear findings for high-grade lesions [9].

The HPV-DNA screening approach enables timely detection of viral infections and precancerous changes, allowing early intervention strategies to prevent progression to cervical cancer [10,11].

Epidemiologic research has led to the classification of HPV types according to their relative risk in cancer development. In particular, HPV types 16, 18, 31, 33, 35, 39, 39, 45, 51, 52, 56, 58, 59, and 68 are recognized as members of the high-risk group. Among these, HPV types 16, 18, 31, 33, 45, 52, and 58 stand out as the main causative agents responsible for more than 90% of cervical cancer patients [12].

The main aim of our study was to make a comparative analysis between the results of CVS and HPV-DNA co-test results and the results obtained from colposcopic biopsy.

## 2. Methods

In this retrospective cohort study, we evaluated 225 patients who underwent primary CVS followed by secondary CVS/colposcopic biopsy and pathology diagnosis at our hospital. The evaluation comprises cases diagnosed at the university hospital of Canakkale Onsekiz Mart in Canakkale, Turkey between 2014 and 2022. Patients before 2014 were not included in the study because their data could not be accessed. The presence of HPV-DNA tests was also sought in these cases, but HPV-DNA results were available in 186 cases. Cases under 18 years of age were not included in the study. No cases of conization following abnormal CVS and HPV-DNA test results were recorded in our department. The comprehensive analysis included electronic screening of cervical cytology and colposcopic biopsy results as well as HPV-DNA analysis results, all documented in our medical record system.

Cytologic examinations were performed on smear samples utilizing both Papanicolaou (PAP) staining techniques, with 59.2% utilizing the conventional smear method and 40.8% employing liquid-based cytology (LBC). Colposcopic biopsies underwent classical hematoxylin and eosin (H&E) evaluation. The re-evaluation and comparative analysis of biopsy and smear results were conducted, with smear samples assessed according to the Bethesda 2022 system [13].

Exclusion criteria for the study included patients who did not undergo an initial cervicovaginal smear and secondary smear/colposcopic biopsy or who did not have a biopsy report in our department following abnormal smear results and who did not undergo HPV-DNA co-testing. These criteria ensured a focused review of cases with complete diagnostic data for a comprehensive understanding of screening results.

Ethical considerations were paramount in the study’s design and execution. The research adhered to the principles outlined in the Helsinki Declaration of 1975, as revised in 2000, and was in compliance with the committee on human experimentation, whether institutional or regional. The study received ethical approval from the relevant ethics committee, and the approval certificate is annexed to this communication. The approval was granted on the 22nd of March 2022, reaffirming the commitment to ethical standards and guidelines in the conduct of this research.

### Statistical Method

The SPSS 25.0 package was used for statistical analysis of the data. Categorical measurements were summarized as number and percentage, and continuous measurements were summarized as mean and standard deviation (median and minimum-maximum where necessary).

The agreement between the diagnoses was evaluated by kappa concordance analysis. The correlation between the diagnoses was evaluated by intraclass correlation method. The kappa statistic takes a value between −1 and +1. Positive values of κ indicate that the agreement between raters is higher than the expected agreement by chance, negative values of κ indicate that the agreement between raters is less than expected by chance [14].

Sensitivity, specificity, positive predictive value (PPV), and negative predictive value (NPV) were calculated.

The Kruskal–Wallis test was used to compare the diagnostic groups and age. Statistical significance level was taken as 0.05 in all tests.

## 3. Results

In our retrospective investigation, the age of the participants ranged from 21 to 86 years, with a mean age of 44.8 years and a standard deviation of 12.7. The median age of the study population was 46 years.

CVS samples consisted of conventional smears (59.2%) and LBC (40.8%) material. CVSs were diagnoses as negative for intraepithelial lesions (negative), atypical squamous cells of undetermined significance (ASCUS), low-grade squamous intraepithelial lesion (LSIL), high-grade squamous intraepithelial lesion (HSIL) (2.7%, 44.4%, 41.3%, 11.6%, respectively) (Figure 1). Colposcopic biopsies or second cervicovaginal smears were diagnosed as negative for intraepithelial lesions, ASCUS, CIN1/LSIL, CIN2-3/HSIL, cervical cancer (36%, 4%, 42.2%, 16.4%, 1.3%, respectively) (Figure 1).

Data on the initial diagnosis by CVS test and subsequent colposcopic biopsy/second CVS test results are shown in Figure 2. Microscopic photographs of the sample cases are shown in Figure 3.

When we examined the concordance of the first diagnosis with the second diagnosis in the whole patient group; the kappa value and intraclass correlation coefficient were 0.31 and 0.64 (95% CI 0.54–0.71), respectively. Among the diagnoses, the highest accuracy rate between the second diagnosis and the first diagnosis was CIN1/LSIL (81.1%), while the lowest accuracy rate was found in the negative result in terms of intraepithelial lesion.

ASCUS was detected in 93.8% of the patients in whom no intraepithelial lesion was detected by CVS and in whom smears were taken for the second time. CIN2/HSIL and CIN3/HSIL were detected in 30% and 2.7% of the patients who were diagnosed with ASCUS by CVS and who had positive HPV-DNA test at the same time.

Most of the patients diagnosed with HSIL by smear were also diagnosed by biopsy (88.4%). There were two cases in which invasive cancer was detected on biopsy after a negative smear test.

With the simultaneous HPV-DNA test, 186 patients were examined, and HPV-DNA was found to be positive in half of the patients (*n* = 92, 49.4%). HPV DNA types 6, 16, 17, 18, 25, 81 were observed (Figure 4) and the most common types were type 16 and type 18 (64.1% and 14.1%, respectively).

In our investigation, the sensitivity and specificity of HPV testing for detecting CIN1/LSIL were determined to be 87.5% and 53.3%. In comparison, the corresponding figures for low-grade cytology testing were 73.5% and 71.7%, respectively. The sensitivity and specificity of HPV testing for detecting CIN2-3/HSIL were determined to be 83.87% and 58.49%, respectively, in biopsy. In comparison, the corresponding figures for high-grade cytology testing were 63.6% and 52.4%, respectively. The PPV for CIN2-3/HSIL was 94.9%, and the NPV for CIN2-3/HSIL was 28.26% in biopsy, while the PPV for CIN2-3/HSIL was 70.3%, and the NPV for CIN 2-3/HSIL was 75.6% in CVS (Table 1).

Regarding the detection of CIN2-3/HSIL, the sensitivity and specificity of HPV testing were found to be 83.87% and 58.49%, respectively, in biopsy. On the other hand, the sensitivity and specificity of cytology testing for CIN2-3/HSIL were 63.6% and 52.4%, respectively. The PPV for CIN2-3/HSIL was 94.9%, and the NPV for CIN2-3/HSIL was 28.26% in biopsy, while the PPV for CIN2-3/HSIL was 15.2%, and the NPV for CIN2-3/HSIL was 91.4% in CVS (Table 1 and Figure 5).

HPV-DNA was positive in one (20%) of the cases with negative results in the first CVS diagnosis, while HPV-DNA was positive in 14 (22.2%) of the cases diagnosed with ASCUS. One case (100%) who was negative in the initial CVS diagnosis and HPV (+) was diagnosed with ASCUS by secondary CVS. In the second examinations of HPV (+) cases diagnosed with ASCUS, 13 (92.8%) were diagnosed with LSIL/CIN1 and one (7.1%) was diagnosed with cancer. Only one (1.5%) of the HPV-negative cases diagnosed as ASCUS at the first diagnosis was re-diagnosed as ASCUS, while 62 (98.4%) received the second diagnosis as negative.

Intraepithelial lesions were detected in the CVS cytology of all patients with positive HPV-DNA test results and there was a statistically significant correlation (*p* = 0.0003). HPV-DNA was positive in most of the cases cytologically diagnosed as CIN1/LSIL and CIN2-3/HSIL (73.6% and 63.6%, respectively). In the cytologic examination of HPV-DNA negative cases, ASCUS was the most common diagnosis (67%). CVS CIN1/LSIL diagnosis was significantly correlated with HPV 16 (*p* = 0.0001).

CIN1/LSIL in 91.1% of HPV 16 positive cases and CIN2-3/HSIL in 40.9% of HPV 18 positive cases were significantly correlated (Fisher’s exact test, *p* = 0.006).

Regarding the correlation between cervical biopsy and HPV, intraepithelial lesions were detected by biopsy in most of the HPV positive cases, but no pathology was observed in five cases (7.9%). There was a significant correlation between HPV test and histopathologic result of cervical biopsy (*p* = 0.0002).

CIN2-3/HSIL diagnosis by CVS became more frequent with increasing age (51.1 ± 10.6), although no statistically significant correlation was found (*p* = 0.058). CIN2-3/HSIL and cancer diagnosed by cervical biopsy were significantly more common at older ages (*p* = 0.001).

Among women over 40 years of age, 25.7% had ASCUS, 27.1% had CIN1/LSIL and 9.7% had CIN2-3/HSIL. Compared to women under 40 years of age, significantly more intraepithelial lesions were diagnosed (*p* < 0.005). The proportion of patients with HPV subtypes other than HPV16 and HPV18 was higher in older patients than in younger patients (84.6% vs. 51.4%, *p* = 0.037).

## 4. Discussion

Within the framework of our study, a cohort comprising 225 cases underwent dual screening with co-test cervicovaginal smear (CVS) and human papillomavirus DNA (HPV-DNA) testing, followed by subsequent colposcopic biopsy and diagnostic evaluation in the pathology laboratory.

Noteworthy is our alignment with the observations made by Aydogan et al. [15], whereby our study echoed the significantly high sensitivity in detecting intraepithelial lesions through cervical smears, regardless of whether the liquid-based cytologic or conventional cytologic method was employed. This consistency in findings reaffirms the efficacy of the conventional method of cervical cytology in the identification of intraepithelial lesions, validating its continued relevance in contemporary diagnostic practices.

It is imperative to acknowledge that while colposcopy is a widely employed diagnostic tool, certain limitations persist in its ability to comprehensively detect CIN2-3/HSIL or cervical cancer lesions. This assertion aligns with the findings reported by Zuchaa et al., who documented a sensitivity of 66.2% for CIN2/HSIL lesions when comparing colposcopy to the gold standard of at least three biopsies of the cervix. Despite these recognized limitations, colposcopy remains a primary diagnostic modality for cervical dysplasia, emphasizing the need for a nuanced understanding of its strengths and weaknesses within the broader context of cervical health assessment [15].

The findings from our study contribute to the ongoing discourse surrounding cervical cancer screening methodologies, particularly by corroborating the effectiveness of conventional cytologic methods in lesion detection. As we navigate the landscape of cervical health diagnostics, it becomes imperative to integrate multiple screening modalities judiciously to ensure a comprehensive and accurate assessment of cervical abnormalities. Future research endeavors should continue to refine our understanding of the strengths and limitations of each diagnostic tool, paving the way for optimized screening protocols and improved patient outcomes.

Nonetheless, it is imperative not to overshadow the merits of liquid-based cytology in relation to conventional smears. Primarily, the utilization of the same collected sample for both HPV testing and cytological examination constitutes a dual advantage, yielding economic efficiencies and mitigating patient discomfort [6]. Additionally, the microscopic scrutiny of liquid-based cytology slides proves to be notably more straightforward, and expeditious results are attainable in comparison to conventional smears [16,17]. Presently, numerous regions still rely on conventional smear cytology, particularly in cases where Italian guidelines mandate primary cytomorphological assessment for women below 30 years of age, eschewing molecular HPV testing. While acknowledged as a cost-effective approach, this practice is lamentable, as it foregoes the inherent advantages associated with liquid-based cytology [5].

Cervicovaginal HPV testing exhibits a sensitivity of 90% in the identification of precancerous conditions. In the general population, the cumulative risk of encountering precancer over a 5-year period subsequent to obtaining a negative HPV test result is less than 0.15%. Individuals with prevailing precancer risks below 4% are advised to undergo repeat HPV testing at intervals of 1, 3, or 5 years, contingent upon the 5-year precancer risk. For those with current precancer risks ranging from 4% to 24%, including individuals with low-grade cytology test outcomes such as ASCUS or CIN1/LSIL coupled with an undetermined-duration positive HPV test, colposcopy is recommended [18]. Conversely, individuals exhibiting current precancer risks within the range of 25% through 59%, such as those with high-grade cytology results denoting ASC-H or CIN2-3/HSIL alongside positive HPV test results, are advised to undergo management procedures involving colposcopy accompanied by biopsy or excisional treatment. This nuanced stratification ensures tailored and evidence-based approaches to the management of cervical precancerous conditions based on individual risk profiles [18].

In our investigation, the diagnostic process entailed either colposcopic biopsy or a second CVS following an initial abnormal CVS result. Comparable to certain studies in the existing literature, a notable association between CVS findings and biopsy results was not observed, as evidenced by a kappa value of 0.31. Among the various diagnostic categories, the highest degree of correlation was noted in cases classified as CIN1/LSIL at 81.1%.

The profound interconnection between HPV and cervical dysplasia is prompting a paradigm shift in conventional screening approaches. The ASCCP advocates for the adoption of co-test screening, involving the combined utilization of HPV-DNA testing and CVS examination, specifically for women within the age range of 30 to 64 years. The co-test methodology exhibits a higher likelihood of detecting abnormal cervical cells or cervical cancer compared to cervicovaginal smear screening in isolation. Furthermore, the co-test offers enhanced feasibility in diverse healthcare settings, making it more accessible than colposcopy [19].

Within the purview of our investigation, the diagnostic trajectory encompassed either a colposcopic biopsy or a secondary CVS following an initial abnormal CVS result. In consonance with certain extant studies, our findings revealed a discernable lack of a robust association between CVS outcomes and biopsy results, as underscored by a kappa value of 0.31. This observation implies a modest agreement between the two diagnostic modalities, emphasizing the need for a nuanced interpretation of screening results. Intriguingly, among the various diagnostic categories, a comparatively higher degree of correlation was discerned in cases classified as CIN1/LSIL at 81.1%, shedding light on potential variations in diagnostic accuracy across distinct pathological classifications.

The pivotal nexus between HPV and cervical dysplasia is reshaping conventional screening paradigms. The American Society for Colposcopy and Cervical Pathology champions the adoption of co-test screening, wherein both HPV-DNA testing and CVS examination are employed in tandem, particularly for women aged 30 to 64 years. This strategic amalgamation is driven by the heightened sensitivity of the co-test methodology in detecting abnormal cervical cells or cervical cancer when compared to isolated cervicovaginal smear screening. Moreover, the co-test approach offers augmented practicality across diverse healthcare settings, rendering it more accessible than colposcopy, thereby aligning with the imperative of enhancing screening reach and efficacy [19].

As our study contributes to the growing body of knowledge in cervical health diagnostics, it underscores the intricate relationship between screening modalities and emphasizes the evolving landscape of cervical dysplasia detection. The insights garnered from this investigation not only contribute to the academic discourse but also hold practical implications for refining screening strategies, ultimately contributing to improved patient outcomes. Future research endeavors should continue to explore innovative approaches and technologies to further enhance the precision and accessibility of cervical cancer screening in diverse clinical contexts.

In our study, similar to the study by Kussaibi H. et al. [20], a high correlation was found between HPV-DNA test and positive cytology and biopsy. There was a significant correlation between HPV-DNA test and cervicovaginal smear result and cervical biopsy (*p* = 0.0003, *p* = 0.0002, respectively). This result emphasizes that HPV-DNA testing should be included in cervical cancer screening methods.

Numerous studies have proven that HPV infection is a precursor virus for the development of cervical precancerous lesions and cervical cancer [21,22]. Moreover, the temporal progression from HPV infection to the manifestation of cervical carcinoma in situ typically spans a duration of approximately 7 to 12 years, as corroborated by various clinical studies. Consequently, the timely identification and prompt management of CIN2/HSIL and higher-grade lesions assume paramount significance in the context of cervical cancer screening efforts [21].

ASCUS represents the most common abnormal diagnostic result encountered in CVS screening tests and is characterized by cell abnormalities that are more prominent than reactive changes but do not reach the threshold of a squamous intraepithelial lesion. Yarandi F. et al. [23] found the rate of detection of higher intraepithelial lesions in CVS screening in patients diagnosed with ASCUS to be 3–10%. In addition, Li X. et al. [24] reported a CIN2/HSIL detection rate of 13.55% in 251 ASCUS cases, while Li SR. et al. [25] reported a CIN2-3/HSIL detection rate of 7.3% in 463 ASCUS cases. Furthermore, the existing literature reports high rates of HPV positivity and the association of lesions with CIN2-3/HSIL among patients with ASCUS, standing at 6.8%, 11%, and 14.3% [24,25,26].

Similarly, in our study, cases with ASCUS identified through CVS and concomitant HPV-DNA positivity exhibited cervical biopsy results indicating CIN2/HSIL in 30% of cases and CIN3/HSIL in 2.7% of cases. This finding suggests that HPV-DNA testing is a test that should be performed in women with ASCUS. Hence, ASCUS represents a lesion warranting thorough evaluation through colposcopy, particularly when HPV-DNA testing yields a positive result.

According to the literature, around 13% of women diagnosed with CIN1/LSIL through CVS screening tests are likely to progress to CIN2-3/HSIL within a 3-year period [13]. In our study, all patients with CIN1/LSIL underwent colposcopy-guided biopsy, and 13 of them were subsequently diagnosed with a higher-stage lesion, specifically CIN2-3/HSIL. This finding aligns with previous reports that indicate such progression rates to range from 12% to 20% [27,28,29]. These results emphasize the crucial role of colposcopic evaluation as an essential step to be undertaken in cases of CIN1/LSIL for accurate diagnosis and appropriate management.

In our study, a statistically significant difference was observed in the diagnosis of intraepithelial lesions between women aged over 40 years and those under 40 years (*p* < 0.005). Specifically, a higher proportion of intraepithelial lesions was detected in women over 40 years of age compared to their younger counterparts. This finding contrasts with the findings reported by Louro et al. [30], where they reported a lower likelihood of detecting clinically significant lesions in the subsequent histologic follow-up of patients aged 40 years and older, as opposed to patients younger than 40 years.

In our study, HPV-DNA testing in combination with CVS cytology yielded a diagnosis of CIN1/LSIL with higher sensitivity and specificity than colposcopic cervical biopsy. In contrast, a study conducted by Wang et al. [21] reported lower sensitivity and specificity between CIN1/LSIL and HPV with cytologic evaluation compared to biopsy. Similar trends were observed in the evaluation of CIN2-3/HSIL in our study, where the sensitivity and specificity between cytology and HPV were lower than those of biopsy (83.87% and 58.49% sensitivity and specificity in biopsy, 63.6% and 52.4% in cytology, respectively).

The severity of cervical lesions may vary according to HPV types. Notably, a progressive escalation in cervical lesion grades is observed with higher viral loads of HPV16 and HPV18, and a robust correlation exists between these viral loads and the severity of the cervical lesions [21]. Depuydt et al. [31] further bolstered this concept, suggesting that the pathological changes observed in CIN3/HSIL may be attributed to a steady increase in the specific HPV type and viral load. Consequently, detection of specific HPV load can be used to predict the risk of CIN3/HSIL occurrence.

Another study emphasized that high-risk HPV types other than HPV16 and 18 can also predict high-risk lesions such as CIN2/HSIL and CIN3/HSIL [32]. In our study, we identified a noteworthy association between HPV 16 and CIN1/LSIL, and between HPV 18 and CIN2-3/HSIL. HPV16/18 viral load increased in proportion to the severity of cervical lesions (*p* = 0.006).

While our study provided valuable insights, it is imperative to acknowledge certain limitations that may influence the interpretation of results. A prospective study design, incorporating a larger sample size, would have bolstered the robustness, reliability, and generalizability of our findings.

## 5. Conclusions

Despite these limitations, our study revealed that the conventional smear technique, while considered older, remained a reliable tool in the detection of high-grade cervical lesions. This underscores the importance of continuous patient follow-up, recognizing that diagnostic outcomes may evolve upon biopsy, necessitating vigilant monitoring for accurate clinical management.

Furthermore, the incorporation of HPV-DNA studies in all cytologic examinations was deemed indispensable, particularly in specific patient subsets. This precautionary measure aimed to avoid unnecessary interventions and contribute to the precision of diagnostic assessments. The accurate detection of cervical lesions through this comprehensive approach holds significance not only in preventing unwarranted surgical procedures but also in empowering clinicians to devise tailored and appropriate follow-up strategies for improved patient management.

## Figures and Tables

**Figure 1 diagnostics-14-00611-f001:**
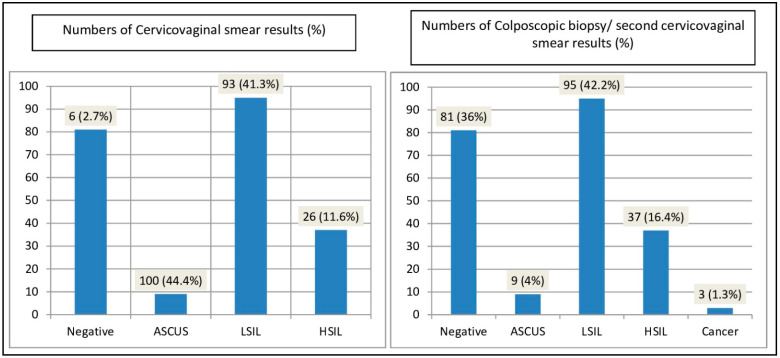
Histological results for patients undergoing CVS and colposcopic biopsy.

**Figure 2 diagnostics-14-00611-f002:**
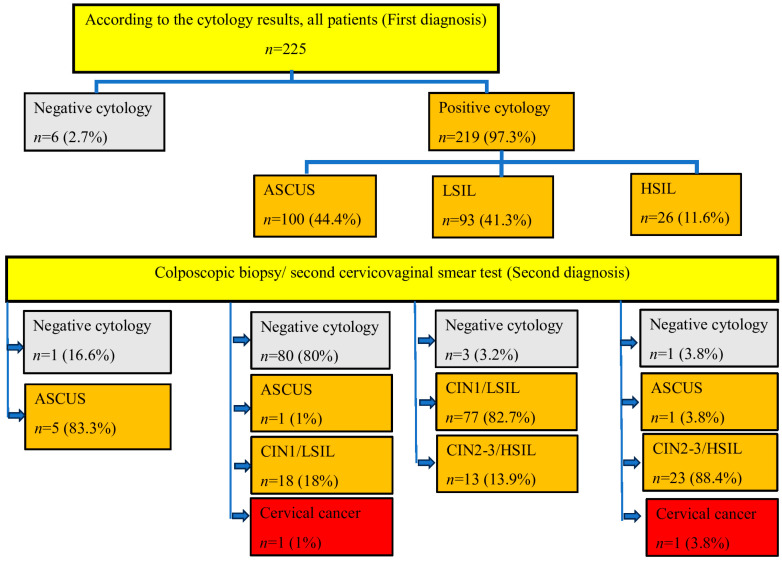
Flowchart of the study.

**Figure 3 diagnostics-14-00611-f003:**
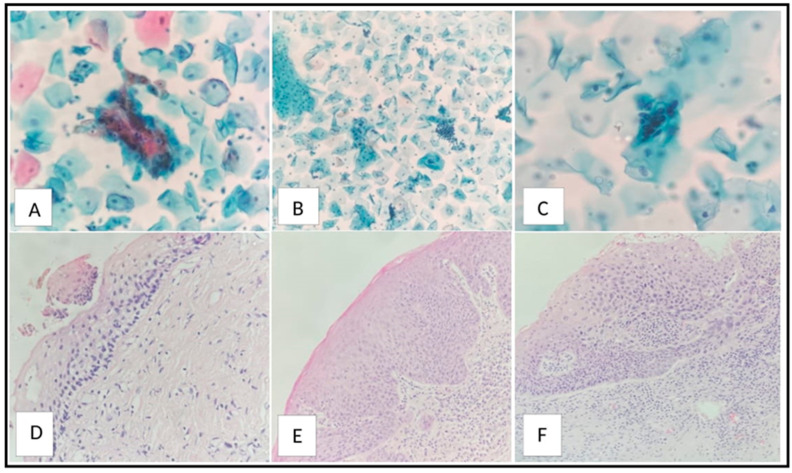
Microscopic photo of a case diagnosed as ASCUS by PAP smear (PAP stain, ×400) (**A**), and CIN1/LSIL by cervical colposcopic biopsy (hematoxylin and eosin stain, ×200) (**D**). Photomicrograph of a case diagnosed as CIN1/LSIL by PAP smear (PAP stain, ×200) (**B**), and CIN2/HSIL by cervical colposcopic biopsy (hematoxylin and eosin stain, ×200) (**E**). Photomicrograph of a case diagnosed as CIN3/HSIL by PAP smear (PAP stain, ×200) (**C**), and CIN3/HSIL by cervical colposcopic biopsy (hematoxylin and eosin stain, ×200) (**F**).

**Figure 4 diagnostics-14-00611-f004:**
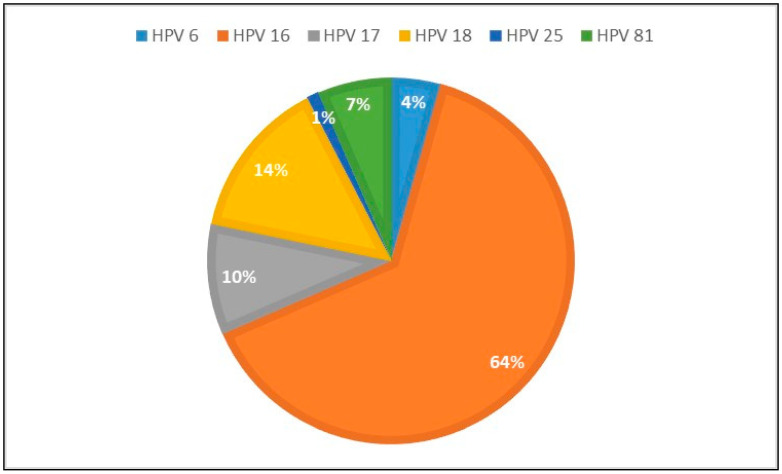
Graph of HPV types of cases according to HPV DNA test results.

**Figure 5 diagnostics-14-00611-f005:**
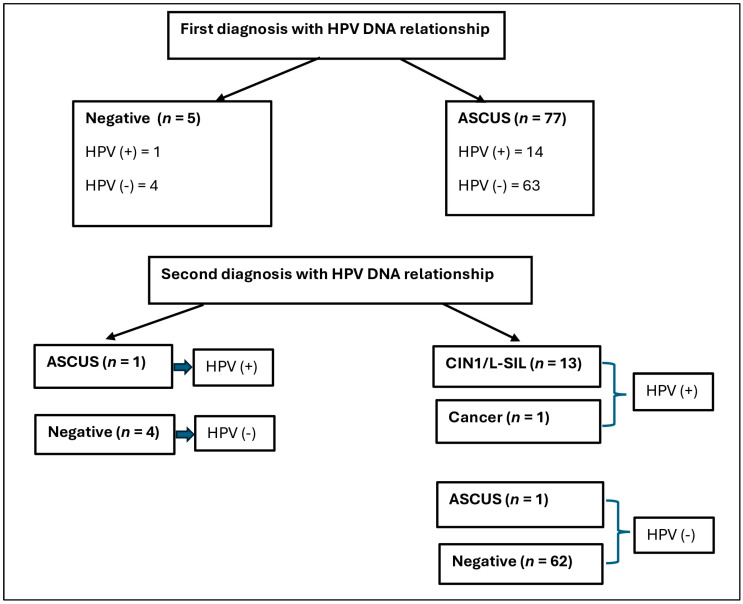
Relationship with HPV-DNA and first/second diagnosis.

**Table 1 diagnostics-14-00611-t001:** Correlation between HPV results (positive) and colposcopic biopsy/Pap smear results.

	Sensitivity	Specificity	PPV	NPV
Smear results				
LSIL	73.5%	71.7%	70.3%	75.6%
HSIL	63.6%	52.4%	15.2%	91.4%
Biopsy results				
CIN1/LSIL	87.50%	53.30%	7.61%	98.98%
CIN2-3/HSIL	83.87%	58.49%	94.9%	28.26%

Abbreviations: PPV: positive predictive value; NPV: negative predictive value; HSIL: high-grade squamous intraepithelial lesions; LSIL: low-grade squamous intraepithelial lesions; CIN 2: cervical intraepithelial neoplasia 2; CIN 3: cervical intraepithelial neoplasia 3.

## Data Availability

The datasets used and analyzed during the current study are available from the corresponding author on reasonable request.

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
