# Peer review of "Assessment of Cervicovaginal Smear and HPV DNA Co-Test for Cervical Cancer Screening: Implications for Diagnosis and Follow-Up Strategies"

_diagnostics, 2024, doi:10.3390/diagnostics14060611_

Round 1

Reviewer 1 Report (Previous Reviewer 1)

Comments and Suggestions for Authors

Thank you for responding to my requests, especially for adding Figure 5. However, I cannot understand the meaning of the number in the left graph of Figure 1, namely 6 to over 80, 100 to less than 10, and so on.

Author Response

Thank you for your comment. You are so right, the figure 1 was wrong I realized that. I have changed this figure.

Reviewer 2 Report (Previous Reviewer 3)

Comments and Suggestions for Authors

The article is generally well-written and further improved. Limitations should move from the conclusions section to the discussion section. Otherwise, happy with the article.

Author Response

I have changed the conclusion part.

This manuscript is a resubmission of an earlier submission. The following is a list of the peer review reports and author responses from that submission.

Round 1

Reviewer 1 Report

Comments and Suggestions for Authors

I think this study is meaningful, because the group of patients with false negative by cytology may include the patients with HSIL or carcinoma. So, I have some requests as follows.

1: I think in this study it was most important to detect the patients who were diagnosed finally LSIL, HSIL or carcinoma, though diagnosed negative or ASCUS by the cytology. So, I want you to show the relationship between the results of HPV-DNA and the final diagnosis by biopsy in the 106 patients, including 6 negative and 100 ASCUS. For example, XX patients with posivite HPV-DNA and YY patients with negative are divided and the each number of the classification by biopsy is compared in a new table.

2: I think that in Figure 1, the values of bar graph in the left, including negative, ASCUS, and HSIL, may be mistaken.

Author Response

1: I think in this study it was most important to detect the patients who were diagnosed finally LSIL, HSIL or carcinoma, though diagnosed negative or ASCUS by the cytology. So, I want you to show the relationship between the results of HPV-DNA and the final diagnosis by biopsy in the 106 patients, including 6 negative and 100 ASCUS. For example, XX patients with posivite HPV-DNA and YY patients with negative are divided and the each number of the classification by biopsy is compared in a new table.

Response 1: 

We can add figure 5 and the examination of the figure 5;

HPV-DNA was positive in 1 (20%) of the cases with negative results in the first CVS diagnosis, while HPV-DNA was positive in 14 (22.2%) of the cases diagnosed with ASCUS. One case (100%) who was negative in the initial CVS diagnosis and HPV (+) was diagnosed with ASCUS by secondary CVS. In the second examinations of HPV (+) cases diagnosed with ASCUS, 13 (92.8%) were diagnosed with LSIL/CIN1 and 1 (7.1%) was diagnosed with cancer. Only 1 (1.5%) of the HPV-negative cases diagnosed as ASCUS at the first diagnosis was re-diagnosed as ASCUS, while 62 (98.4%) received the second diagnosis as negative.

Legend of figure 5:

Figure 5: Relationship with HPV-DNA and first/second diagnosis.

2: I think that in Figure 1, the values of bar graph in the left, including negative, ASCUS, and HSIL, may be mistaken.

Response 2: I have checked it, and I am sure it is correct.

Reviewer 2 Report

Comments and Suggestions for Authors

This study evaluated the fole of cervical cytology and HPV test in CC screening. The introduction is poorly written and incidence rates of cervical precancerous lesions and cervical cancer in authors' country are not presented. The study population is not properly defined as it is not clear why the study period between 2014 and 2022 was selected. The role of cervical cytology and HPV testing in CC screening is well established with pros and cons of both approaches well understood. It is not clear what were the indications for cotesting? The patient exclusion is also not clearly presented.

Author Response

Reviewer 2: This study evaluated the fole of cervical cytology and HPV test in CC screening. The introduction is poorly written and incidence rates of cervical precancerous lesions and cervical cancer in authors' country are not presented. The study population is not properly defined as it is not clear why the study period between 2014 and 2022 was selected. The role of cervical cytology and HPV testing in CC screening is well established with pros and cons of both approaches well understood. It is not clear what were the indications for cotesting? The patient exclusion is also not clearly presented.

-We have written the introduction part again;

"Cervical cancer, the fourth most common cancer in women and the second leading cause of mortality among malignancies, resulted in 528,000 cases and 311,000 deaths worldwide in 2018. This epidemiologic profile underscores the cancer's continued status as a major public health problem1-3.

There are significant regional variations in incidence depending on the status of Human Papillomavirus (HPV) vaccination and the availability of cervicovaginal smear (CVS) screening4,5.

In 2012, the American Society for Colposcopy and Cervical Pathology (ASCCP) revised a comprehensive, evidence-based consensus on the management of abnormal cervical cytology, cervical intraepithelial neoplasia and adenocarcinoma. The ASCCP recommends screening with Co-Test HPV DNA and smear for women aged 30-64 years. This strategy not only achieves the highest sensitivity for cancer detection, but also exhibits the highest negative predictive value by providing longer intervals between negative test results6,7.

In Turkish studies, ASCUS is the most common abnormal smear result, sparking controversy in management. Research heavily focuses on ASCUS and LSIL patients, with ASCUS cases benefiting from comprehensive evaluation, preferably with HPV testing, especially with liquid-based cytology. Conversely, colposcopy is recommended for LSIL patients, particularly with high HPV positivity. However, cytologic follow-up has drawbacks like missed CIN2/3 lesions, potential delays in cancer diagnosis, repeated testing, patient non-compliance, and increased anxiety8.

Aydogan et al. conducted a study assessing cervical cytology using conventional methods. They found a sensitivity of 70.8% and specificity of 62.2% (with positive predictive value (PPV) and negative predictive value (NPV) both at 66.7%) for detecting low-grade lesions. Similarly, for high-grade lesions, the sensitivity was 72.4% and specificity was 86.0% (with PPV at 70.0% and NPV at 87.3%). Among patients with HSIL/ASC-H, 73.5% were diagnosed with high-grade lesions following colposcopy. Moreover, CIN+2 lesions were identified in 26.9% of patients with ASCUS and high-risk HPV. The results of smear-colposcopy-guided biopsy closely correlated with smear findings for high-grade lesions9.

The HPV-DNA screening approach enables timely detection of viral infections and precancerous changes, allowing early intervention strategies to prevent progression to cervical cancer 10,11.

Epidemiologic research has led to the classification of HPV types according to their relative risk in cancer development. In particular, HPV types 16, 18, 31, 33, 35, 39, 39, 45, 51, 52, 56, 58, 59 and 68 are recognized as members of the high-risk group. Among these, HPV types 16, 18, 31, 33, 45, 52 and 58 stand out as the main causative agents responsible for more than 90% of cervical cancer patients12.

The main aim of our study was to make a comparative analysis between the results of CVS and HPV-DNA cotest results and the results obtained from colposcopic biopsy."

-We can change the first paragraph of the method;

"In this retrospective cohort study, 225 cases diagnosed in our hospital with cervicovaginal smear followed by secondary cervicovaginal smear/colposcopic biopsy between 2014 and 2022 for which data were available were included. Patients before 2014 were not included in the study because their data could not be accessed. The presence of HPV-DNA test was also sought in these cases, but HPV-DNA results were available in 186 cases. Cases under 18 years of age were not included in the study. No cases of conization following abnormal CVS and HPV-DNA test results were recorded in our department. The comprehensive analysis included electronic screening of cervical cytology and colposcopic biopsy results as well as HPV-DNA analysis results, all documented in our medical record system."

-We can change the third paragpaph of the method;

"Exclusion criteria for the study included patients who did not undergo an initial cervicovaginal smear and secondary smear/colposcopic biopsy or who did not have a biopsy report in our department following abnormal smear results with cotest HPV-DNA. These criteria ensured a focused review of cases with complete diagnostic data for a comprehensive understanding of screening results."

Reviewer 3 Report

Comments and Suggestions for Authors

This is an original retrospective research study aiming to evaluate the performance of the conventional pap smear and the addition of HPV-DNA typing. The study revealed that the conventional smear technique, while considered older, remains a reliable tool in the detection of high-grade cervical lesions. The incorporation of HPV-DNA studies in most cytologic examinations further increases sensitivity and specificity. Introduction is adequate, study design and statistical analysis are valid, results are well presented and discussed in detail. Limitations are also included. I think the conclusions section could be reduced to maximum 2 paragraphs and part of the information there could be erased or moved to the discussion section. Otherwise the manuscript has the potential of publication.

Author Response

Reviewer 3: This is an original retrospective research study aiming to evaluate the performance of the conventional pap smear and the addition of HPV-DNA typing. The study revealed that the conventional smear technique, while considered older, remains a reliable tool in the detection of high-grade cervical lesions. The incorporation of HPV-DNA studies in most cytologic examinations further increases sensitivity and specificity. Introduction is adequate, study design and statistical analysis are valid, results are well presented and discussed in detail. Limitations are also included. I think the conclusions section could be reduced to maximum 2 paragraphs and part of the information there could be erased or moved to the discussion section. Otherwise the manuscript has the potential of publication.

Response 3: We can write conclusion part as;

"While our study provided valuable insights, it is imperative to acknowledge certain limitations that may influence the interpretation of results. A prospective study design, incorporating a larger sample size, would have bolstered the robustness, relia-bility, and generalizability of our findings.
Despite these limitations, our study revealed that the conventional smear tech-nique, while considered older, remained a reliable tool in the detection of high-grade cervical lesions. This underscores the importance of continuous patient follow-up, recognizing that diagnostic outcomes may evolve upon biopsy, necessitating vigilant monitoring for accurate clinical management.

Furthermore, the incorporation of HPV-DNA studies in all cytologic examinations was deemed indispensable, particularly in specific patient subsets. This precautionary measure aimed to avoid unnecessary interventions and contribute to the precision of diagnostic assessments. The accurate detection of cervical lesions through this com-prehensive approach holds significance not only in preventing unwarranted surgical procedures but also in empowering clinicians to devise tailored and appropriate fol-low-up strategies for improved patient management."